# A Novel Time–Frequency Feature Fusion Approach for Robust Fault Detection in a Marine Main Engine

Hong Je-Gal [1], Seung-Jin Lee [1], Jeong-Hyun Yoon [1], Hyun-Suk Lee [1], Jung-Hee Yang [2] and Sewon Kim [1,*]

[1] Department of Intelligent Mechatronics Engineering, Sejong University, Seoul 05006, Republic of Korea; jagrhong@sju.ac.kr (H.J.-G.); lsjin6768@sju.ac.kr (S.-J.L.); dbswjdgus9@gmail.com (J.-H.Y.); hyunsuk@sejong.ac.kr (H.-S.L.)

[2] Smart Ship Solution Department, Hanwha Ocean Co., Ltd., Seoul 04527, Republic of Korea; rell1010@hanwha.com

[*] Correspondence: sewonkim@sejong.ac.kr

**Abstract:** Ensuring operational reliability in machinery requires accurate fault detection. While time-domain vibration pulsation signals are intuitive for pattern recognition and feature extraction, downsampling can reduce analytical complexity, but may result in low-precision data, affecting fault detection performance. To address this, we propose time–frequency feature fusion, combining information from both the time and frequency domains for fault detection. Our approach transforms vibrational pulse data into instantaneous revolutions per minute (RPM) and employs statistical analysis for the time-domain features. For the frequency-domain features, we use the combined method of empirical mode decomposition and independent component analysis (EMD-ICA), along with the Wigner bispectrum method to capture the nonlinear characteristics and phase conjugation. Using a deep neural network (DNN), we classify the anomaly states, demonstrating the effectiveness and versatility of our approach in detecting anomalies and improving diagnostic precision. Compared to using time or frequency features alone, our time–frequency feature fusion model achieves higher accuracy, with 100% accuracy at lower downsampling rates and 96.3% accuracy at a downsampling rate of $100\times$.

**Keywords:** fault detection; marine main engine; deep neural network; predictive maintenance; time–frequency feature fusion

## 1. Introduction

### 1.1. Background

The maritime industry plays a crucial role in international trade, with shipping being responsible for transporting over 80% of the global trade in goods [1]. Internal combustion engines (also referred to as main engines in vessels), have been widely used for decades as the primary propulsion system in marine vessels. This choice is due to their cost effectiveness and their ability to generate sufficient power for carrying heavy loads. Despite concerns about their environmental impact stemming from the use of low-quality fuel oil, a significant percentage of vessels continue to rely on internal combustion engines.

However, like any mechanical system, main engines are susceptible to faults that can result in expensive repairs, navigation delays, and even marine accidents. Therefore, the development of effective condition monitoring and fault detection techniques is crucial to mitigate these issues. With the increasing prominence of autonomous vessels, fault detection technology has gained significant attention. As maritime autonomous surface ships (MASS) aim to operate without human intervention, autonomous systems must be capable of detecting and managing faults in main engines [2]. In fact, according to a study by Felski et al. [3], predictive maintenance technology is identified as the primary technological obstacle to achieving full autonomy, surpassing even navigation solutions, which have already been highly automated, such as tracking control systems.

While fault detection and maintenance systems heavily rely on sailors and engineers aboard vessels, fault diagnosis technology in autonomous vessels, particularly main engine fault detection, assumes paramount importance. A vessel cannot navigate without a functioning main engine, and the fault detection technology developed for main engines can also be extended to other rotational equipment, including generator engines, pumps, and oil purifiers, whose normal operations are crucial to normal sailing. Despite the recognized significance of anomaly detection technology, it is worth noting that, currently, only a mere 2% of classed ships utilize condition-based monitoring, as highlighted by Jimenez et al. [4].

Among various approaches to condition monitoring, there is a growing interest in the utilization of vibration data to detect faults in maritime diesel engines. Vibration signals provide valuable insights into the health of the engine, as they can capture minute changes in the operating conditions and identify abnormal patterns or frequencies. With advancements in signal processing and machine learning techniques, vibration data analysis has emerged as a powerful tool for fault diagnosis in the maritime industry.

This manuscript is focused on vibration-based anomaly detection in real-scale ship main engines and presents a two-stage approach, as illustrated in Figure 1.

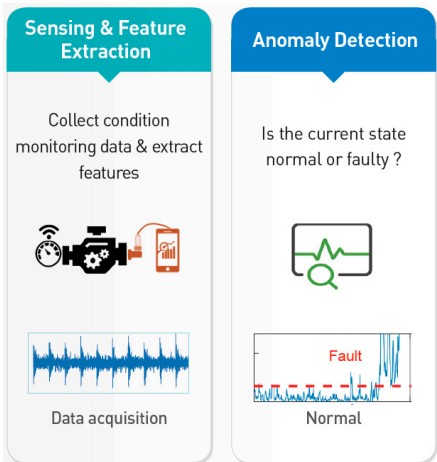

**Figure 1.** Description of the main engine anomaly detection stages.

*1.2. Literature Review*

In this section, we provide a comprehensive review of the existing literature pertaining to fault detection and diagnosis in machinery. Our focus centers on three key themes: acoustic signals, vibration signals, and other data-oriented approaches for fault detection and diagnosis.

1.2.1. Acoustic Signal Source-Based Fault Detection and Diagnosis

Acoustic signal sources are one of the most common data sources for machinery fault detection and diagnosis, thanks to their affordability and accessibility. Several notable studies have made significant contributions to this field. Albarbar et al. [5] proposed a fuel injection condition monitoring model for diesel engines by decomposing acoustic data using independent component analysis (ICA) and extracting fault feature vectors through Wigner–Ville distribution (WVD). Glowacz et al. [6] monitored the condition of induction motors by generating feature vectors using the method of selection of amplitudes of frequency-Multiexpanded (MSAF-20-MULTIEXPANDED) and classified the bearing and stator faults using various classifiers, including the nearest neighbor, nearest mean, and Gaussian mixture model. Yao et al. [7,8] utilized acoustic signals from marine diesel engines to detect abnormal conditions However, despite the advantages, acoustic signals have been reported to exhibit less efficacy as fault detection sources in comparison to vibrational signals. Although acoustic signals offer relative affordability and flexibility, they necessitate

a higher level of sensor maintenance and attachment complexity than vibrational signals [9]. This discrepancy in the maintenance requirements contributes to the complexity of utilizing acoustic signals for fault detection. Conversely, vibrational signals, which are explained in the next subsection, prove to be more sensitive to alterations in the machine's condition. Due to the dynamic vibrational characteristics of machines when faults arise, vibration signal-based fault diagnosis has gained significant prominence [10].

### 1.2.2. Vibration Signal Source-Based Fault Detection and Diagnosis

A vibration sensor is a sensing gadget that traces vibration with a piezoelectric accelerometer. The sensor is used for measuring fluctuating accelerations or speeds, or for normal vibration measurement. Therefore, it has been widely used for machine fault detection or diagnosis tasks. Several notable studies have advanced the field of vibration signal-based fault detection and diagnosis. Li et al. [11] used both vibration signals and lubricating oil wear particle information from marine diesel engines to detect faults using the independent component analysis reference algorithm, effectively detecting fault-related features with expert knowledge. Xi et al. [12] proposed an automatic vibration source extraction and feature visualization technique to detect faults in marine main engines. They employed Stockwell transform to construct a time–frequency reference signal, t-distributed stochastic neighbor embedding (t-SNE) to visualize the fault features, and an extreme learning machine (ELM) classifier to determine the fault presence in the feature vectors. Yan et al. [13] studied a fault diagnosis method for marine blowers using vibration signals, combining ensemble empirical mode decomposition (EEMD), the autoregressive spectrum model, and the correlation coefficient technique. Cheng et al. [14] effectively diagnosed faults in axle-box bearings in high-speed trains by analyzing the non-stationary vibration signals, with a combination of improved EEMD with adaptive noise (IEEMDAN) and complementary EEMD (CEEMD), which breaks down the signals into several intrinsic mode functions. Wang [15] proposed a fault analysis model for a marine engine turbocharger by decomposing the pulsation signals using ensemble empirical mode decomposition and the Teager energy operator. Unlike the previous studies that focused on acoustic fault analysis in marine engines using blind source separation (BSS) to separate the sources, Liu et al. [16] concentrated on applying BSS to vibration signals for decomposition and feature analysis. Li et al. [17] decomposed the feature vectors using independent component analysis (ICA) and short-time Fourier transform, reduced the dimensionality with principal component analysis (PCA), and employed a fuzzy neural network to classify the faults from multi-channel vibration signals. Jing et al. [18] used Fast ICA to decompose the engine vibration data and combined it with a support vector machine (SVM) for engine valve train clearance fault classification. Li et al. [19] proposed a fault recognition model combining ICA and a Factorial Hidden Markov Model for bearing fault detection. Wang et al. [20] introduced a novel model for bearing fault detection by combining generalized composite multiscale weighted permutation entropy (GCMWPE), supervised Isomap (S-Isomap), and SVM. Qu et al. [21] conducted an anomaly detection study with data collected from the vibration acceleration signal on the cylinder head of a four-stroke diesel engine. Their work combined echo state networks and auto-encoders, yielding superior fault detection performance compared to support vector regression and independent auto-encoder models.

Both vibrational and acoustic sensors are susceptible to ambient noise that surrounds the machinery. Many studies aim to extract the original source from the raw signal using algorithms like ICA and EMD. However, when it comes to detecting faults in marine machinery through acoustic sensing, there are challenges. These challenges arise from internal engine noises, vibrations, and additional irregular sounds from surrounding machinery in the engine room. This makes acoustic sensing less effective. On the other hand, vibration sensors, known for their sensitivity and precision, can reliably capture the target signals.

There were also approaches that focused on fault detection by monitoring the engine's rotational movement abnormalities. And since vibrational sensors are commonly

attached to the engine's flywheel or chain box, both located at the end of the engine, they provide better information about the engine's rotation compared to acoustic sensors. Our approach centers on identifying unusual engine revolution patterns. These patterns were derived from vibrational pulse signals, which clarifies why we chose vibrational sensors for this task.

### 1.2.3. Other Data-Oriented Approaches for Fault Detection and Diagnosis

Within this subsection, we briefly introduce alternative methodologies that harness an array of sensing signals emanating from rotating machinery to accomplish the task of fault detection. This exploration extends beyond the realms of acoustic and vibrational signals. Numerous studies have contributed to fault detection and diagnosis using various signal sources. Researchers have explored fault detection techniques for rotating machinery, focusing on issues such as bearing partial rub and looseness [22], and irregular operation modes [6,20,23,24]. Jana et al. [25] developed a fault detection model for real-time accelerometer sensors based on a convolutional neural network and a convolutional autoencoder. Silva et al. [26] proposed a waveforms fault detection model using wavelet transform and neural networks, which were trained with oscillographic data. Utilizing diesel engine sensor data, like lubrication oil filter pressure and temperature, Khelil et al. [27] detected faults in the lubrication system of marine engines using neural networks. Kowalski et al. [28] revealed a multi-class fault diagnosis ensemble learning model trained on feature vectors from marine engines, such as engine load, speed and systems of cooling, fueling, and lubrication. Lima et al. [29] employed short-time Fourier transform (STFT) to extract the main harmonics of the phase current to detect high impedance faults in motors. Moschopoulos et al. [30] combined vibration and acoustic signals from bearings to predict journal bearing performance using machine learning algorithms, such as k-nearest, decision tree, random forest, and gradient tree boosting algorithms. Brandsæter et al. [31] utilized diesel generator data, including engine speed, lubricant oil pressure and temperature, power and bearing temperature, to detect anomalies in real-time by using the method named LSTM-based VAE (variational autoencoder) with image thresholding.

Kim et al. [32] adopted an explainable anomaly detection framework using maritime main engine sensor data, such as fuel oil temperature and pressure, scavenging air temperature and exhaust gas temperature, and proposed a method analyzing voyage data from container ships. Lazakis et al. [33] had an SVM trained with vessel voyage noon reports to monitor the condition of marine diesel generator engines. Lazakis et al. [34] classified critical marine main engine system parameters (e.g., exhaust gas temperatures) using fault tree analysis (FTA) and failure mode and effects analysis (FMEA), and predicted the diesel engine's exhaust gas temperature abnormalities using an artificial neural network (ANN). Ou et al. [35] proposed the identification and reconstruction of anomalous sensing data for combustion in marine diesel engines by using cylinder pressure data and the corresponding crankshaft angle data.

To achieve accurate anomaly detection from signal data, striking the right balance on data precision was vital. Collecting data with excessive precision could burden the limited storage and processing capabilities without providing significant benefits. On the other hand, insufficient data precision may hinder anomaly characteristics extraction, impeding model training and utilization. Low data precision could occur due to sensor failures or bandwidth limitations during data collection. Hence, developing an anomaly detection model that effectively captured anomaly characteristics, even in scenarios with relatively low data precision, became crucial.

Previous studies on marine engine fault detection, such as the work by Li et al. [36], primarily focused on extracting statistical features in the time domain, such as the mean, variance, kurtosis, and asymmetry of the instantaneous revolutions per minute (RPM) data, to differentiate between normal and faulty states and to achieve satisfactory fault detection performance. However, these statistical features in the time domain might be compromised and partly lost when measuring precision decreases. Downsampling

techniques are commonly applied to reduce feature complexity and data size, but they also result in reduced precision of the original data's features. Consequently, existing studies might encounter limitations when dealing with data with compromised precision.

*1.3. Contribution*

To address this challenge, our study went beyond solely relying on time-domain features for training the classification network. Instead, we introduced a novel approach that integrated frequency features alongside time-domain features. By integrating information from both domains, our method remained robust and effective, even in the face of the sensing signal's low precision.

Furthermore, while several studies [11,12,18,24,28], among others, have conducted thorough experiments to detect or diagnose faults in marine diesel engines, few have measured or observed real signals when the main engine operates under normal sailing. The data used in this study was gathered during the onboard quay trial immediately after the test vessel was launched from the shipyard. This unique aspect sets our study apart from other experimental works, providing a valuable perspective on fault detection in real-world operational conditions.

In this paper, the authors proposed the time–frequency feature fusion method that combines feature information extracted from both the time and frequency domains. This approach aimed at maintaining stable anomaly detection performance, even in environments with low data precision. Through a series of experiments, the authors demonstrated the effectiveness of this method for anomaly detection in low-data-precision settings.

In the following sections of this paper, we provide a comprehensive overview of the methodologies and experimental results pertaining to fault detection in machinery. Section 2 presents an exposition on the measurement methods utilized to acquire the actual measurement data used in this study, along with a detailed explanation of the obtained data. In Section 3, we discuss the data processing techniques applied to the acquired data and introduced the deep neural network (DNN) utilized for fault diagnosis based on the processed data. Sections 4 and 5 delve into the methodologies employed for data analysis in the time and frequency domains, respectively, and present the experimental results derived from each analysis. Additionally, in Section 6, we showcase the experimental results from fusing the time–frequency features, which are based on the analysis results obtained from both domains.

## 2. Problem Definition

This study addressed two main objectives: (1) the development of an anomaly detection model that utilizes instantaneous RPM from a vibration encoder, and (2) the identification of relevant feature information for the model.

The research focused on a very large crude oil carrier (VLCC) vessel with a 300,000 deadweight tonnage and a two-stroke diesel engine, specifically the Doosan engine G70ME-C9.2TII model, as depicted in Figures 2 and 3. Figure 3 illustrates the description of the data acquisition site for the abnormal operation test by an encoder. The encoder used in this study was an NI-922, the vibration data acquisition (DAQ) model released by National Instruments. The sensor was installed at the fore side of the engine. The real-scale main engine was utilized to gather vibration data for the fault detection task.

During the onboard quay trial or commissioning test for the newly built vessel at the shipyard, both the normal and fault operation modes were imposed on the main engine (a trial is conducted to ensure that all the machinery is fully functional and ready for sailing before the vessel departs from the shipyard). To simulate engine faults, misfiring of the main engine cylinder was deliberately induced, creating abnormal operating conditions. As a result of this deliberate induction of faults, the measured data obtained during the trial was accurately labeled as either normal or abnormal, allowing for clear separation between the two states for further analysis and fault detection evaluation.

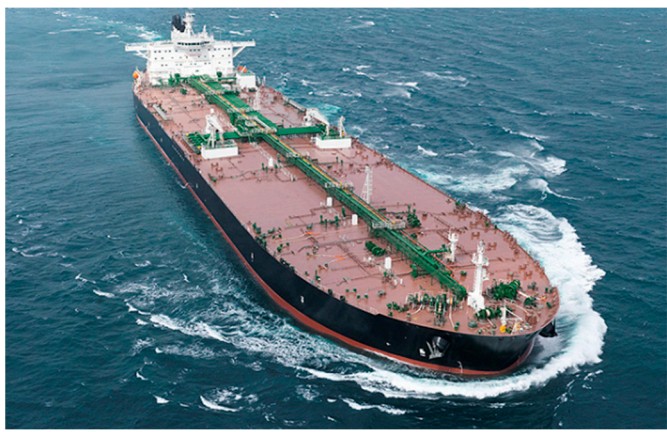

**Figure 2.** Deadweight ton, very large crude carrier.

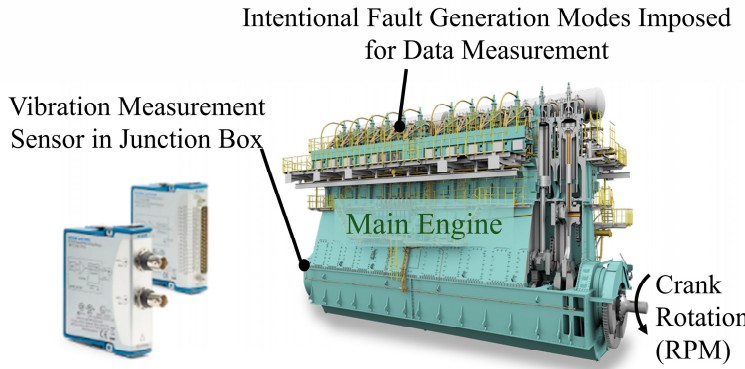

**Figure 3.** Description of the main engine and vibration encoder sensor (NI-922).

Table 1 presents the principal specifications of the main engine used in this study, including the model type, the maximum power output, the number of cylinders, and the normal continuous rating (NCR) at 83 RPM.

**Table 1.** Main engine principal specifications.

| Main Engine Model Type | Max Power (kW) | Number of Cylinders | Normal Continuous Rating (NCR) |
|---|---|---|---|
| Doosan Engine G70ME-C9.2 TII (MAN B&W Licensed) | 29,120 kW | 8 | 83 RPM |

The authors collected a multitude of vibration data from the sensor, which are represented as pulse data, as shown in Figure 4. Although pulse data are not ideal for capturing engine characteristics, they are commonly converted into instantaneous RPM, which provides more suitable information for anomaly detection.

The conversion process involved determining the period of each pulse, considering that every pulse represents a rotation of the engine by 1 degree. To find the pulse period, a reference point, known as the starting point of the gear teeth, was required to divide the connected pulse data. In Figure 4, this reference point is represented by points A and B. By calculating the period of each pulse using the reference point, the instantaneous RPM for each pulse could be obtained using Formula (1).

$$RPM = \frac{60}{T \times 360} \tag{1}$$

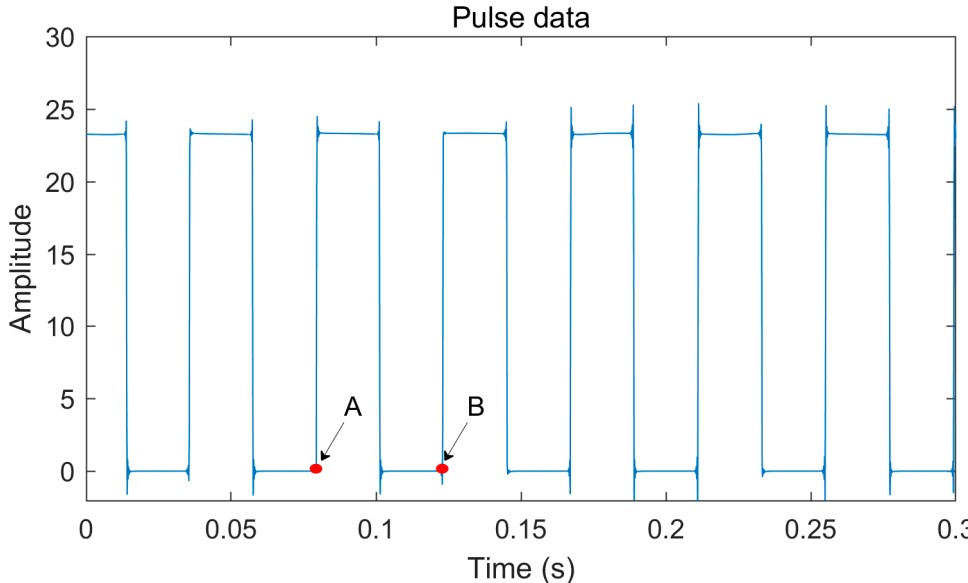

**Figure 4.** Pulse data measured by the encoder. (points A and B are starting points of gear teeth).

In Formula (1), *T* represents the period of each pulse, which corresponds to the duration (in seconds) between points A and B in Figure 4. As *T* represents a 1 degree rotation of the engine, it is multiplied by 360 to convert it into one full rotation and to calculate the instantaneous RPM. The result is illustrated in Figure 5. This data comprised multiple signals with varying amplitudes and frequencies, containing crucial characteristics information. However, since extraneous noise can affect the accuracy of pulse data measurements, the converted instantaneous RPM data may also contain unnecessary information. Hence, it is essential to extract only the relevant features for anomaly detection from the instantaneous RPM data. To achieve this, signal processing techniques in the time domain and frequency domain were explored to extract the data suitable for abnormality detection from the instantaneous RPM data, eliminating unwanted information.

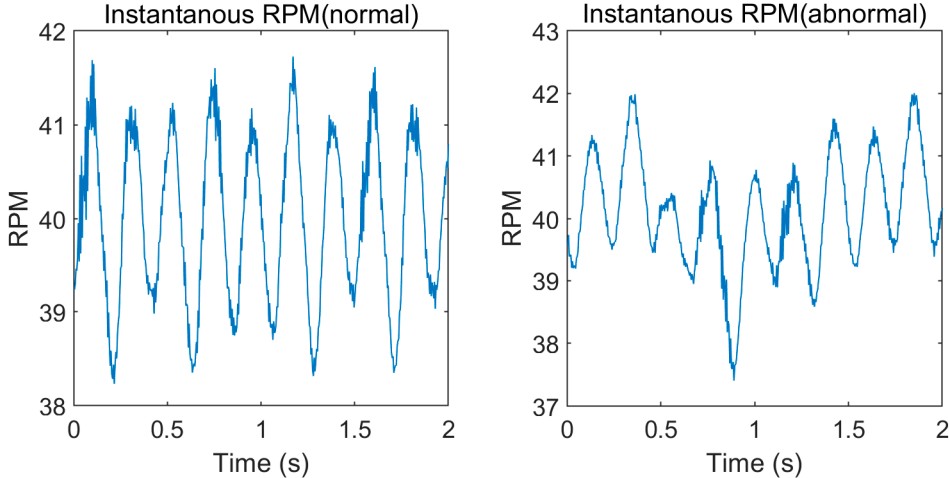

**Figure 5.** Converted instantanous RPM.

## 3. Methodology for Performance Evaluation

This section presents the methodology employed for evaluating the performance of our proposed approach. It encompasses the framework for our procedure, the deep learning model utilized, the data preprocessing techniques, and the performance evaluation metrics employed during the experiments.

Figure 6 illustrates the comprehensive framework for our proposed procedure. The process began with the generation of RPM data using Formula (1). During the data processing stage, we optionally applied data augmentation techniques to enhance the dataset. Next, we performed feature extraction on the dataset in both the time and frequency domains. The extracted features from these domains were then fused together. Subsequently, we employed a deep neural network (DNN) to predict the fault detection using the feature fusion dataset. Additionally, to gain further insights, we generated datasets with varying precision during the data processing stage and implemented separate fault detection DNN models for each individual domain's features.

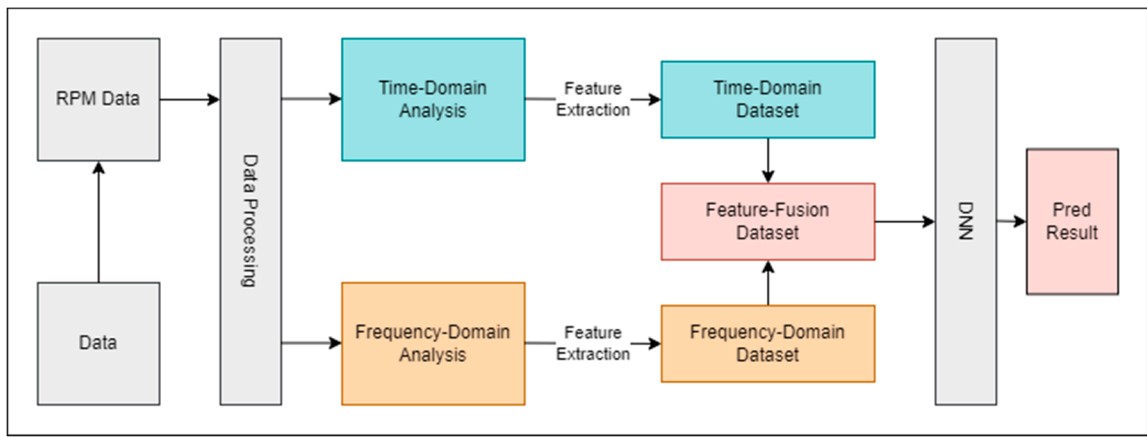

**Figure 6.** Framework.

### 3.1. Deep Neural Network (DNN)

As the measured instantaneous RPM data included both normal and abnormal states, a supervised learning approach was employed for anomaly detection. In this study, a classification approach was adopted to distinguish between normal and abnormal states, with a DNN model serving as the anomaly detection model.

The DNN model consisted of an input layer, a series of hidden layers, and an output layer. The input layer had N nodes, which represent the number of features extracted from the signal data. The output layer had two nodes and utilizes the sigmoid function to classify normal and abnormal states with probabilities. The DNN model had seven hidden layers, and the rectified linear unit (ReLU) function was used as the activation function in each layer. The specific configuration of the model's layers and nodes is outlined in Table 2. The parameters of the model, including the loss function, optimizer, learning rate, and number of training iterations, are specified in Table 3.

**Table 2.** Deep neural network structure.

| Layer | Details |
|---|---|
| Input layer | N nodes |
| First hidden layer | 50 nodes/ReLU |
| Second hidden layer | 80 nodes/ReLU |
| Third hidden layer | 120 nodes/ReLU |
| Fourth hidden layer | 120 nodes/ReLU |
| Fifth hidden layer | 120 nodes/ReLU |
| Sixth hidden layer | 80 nodes/ReLU |
| Seventh hidden layer | 50 nodes/ReLU |
| Output layer | 2 nodes/sigmoid |

**Table 3.** Deep neural network parameters.

| Loss Function | Optimizer | Epoch |
|---|---|---|
| Cross entropy | Adam (learning rate = $1 \times 10^4$) | 4000 |

### 3.2. Data Processing

The raw data used in this study were obtained from the vessel's engine during the onboard quay trial after machinery installation. Due to environmental constraints, the amount of collected data was insufficient for training the DNN adequately. Therefore, to augment the limited dataset, we implemented a sliding window technique, which is commonly employed for the data augmentation of time-series data.

As illustrated in Figure 7, a single time-series data sample was sliced into multiple segments that had overlapping regions. Then, we used each segment as a data sample instead of the original one. By using this technique, the data size within each sample became smaller than the original one. However, it led to an expansion in the total amount of data, while preserving the data's time-series characteristics.

$$\text{Low precision data [n]} = \text{Original data [k} \times \text{n]} \tag{2}$$

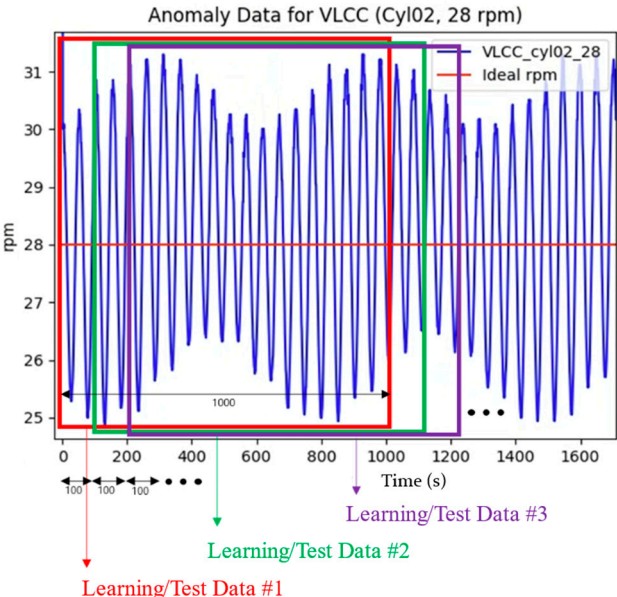

**Figure 7.** Sliding window method.

The goal of this paper was to develop a feature extraction technique that maintains DNN fault detection accuracy, even in a low-precision data environment. To simulate such low-precision data from the given dataset, we artificially lowered the data precision of the original data by downsampling it using Formula (2). In the formula, the variable n represents the index of the data sample, while the variable k determines how much we lowered the data precision compared to the original data, denoted as ($1\times$, $2\times$, $5\times$, $10\times$, $20\times$, $50\times$, $100\times$) with respect to the original data precision. Consequently, the original data were divided into seven groups based on precision, and all the prediction data were converted into pulse to RPM using the RPM calculation process described in Section 2.

### 3.3. Performance Evaluation Metrics

For evaluating the model's performance, accuracy was used as the criterion. A confusion matrix was employed, representing a table with four different combinations of predicted and actual values, as illustrated in Figure 8. True positive (*TP*) refers to the model correctly predicting the true category, while true negative (*TN*) indicates the model

accurately predicting the false category. Both *TP* and *TN* represent correct predictions. False positive (*FP*) occurs when the model predicts the true category, but the actual value is false. False negative (*FN*) arises when the model predicts the false category, but the actual value is true. The accuracy of the model is defined in Formula (3).

$$(Accuracy) = \frac{TP + TN}{TP + FN + FP + TN} \tag{3}$$

| | | Actual values | |
|---|---|---|---|
| | | True | False |
| Predicted Values | True | True Positive | False Positive |
| | False | False Negative | True Negative |

**Figure 8.** Confusion matrix.

## 4. Time Domain

### 4.1. Time-Domain Feature Extraction

To effectively detect anomalies in real-time RPM data using a DNN model, it was crucial to extract features that captured the data's characteristics. Previous studies have focused on extracting features from instantaneous RPM data in the time domain, with statistical features being a widely used approach. When an anomaly occurred in the normal state of a rotating machine, the probability density function of the vibration data changed. As a result, statistical features, such as the maximum, minimum, median, mean, variance, skewness, and kurtosis of the data, extracted from the time domain, exhibited significant variations. These statistical features, which captured the changes in the probability density function, played a pivotal role in distinguishing between abnormal and normal states.

In this study, the authors defined this statistical feature information that changed over time as the "Time Domain Feature Information", since it was extracted from the RPM data that varies with time. However, using the time-domain feature information directly as an input for the DNN model may lead to biased learning, since there are differences in the range of values. To mitigate this bias, a certain range of features was extracted by subtracting the ideal RPM (command speed from the engine governor) from each time-domain feature information. Therefore, the seven time-domain features extracted from the instantaneous RPM data and their corresponding formulas are presented in Table 4.

**Table 4.** Time-domain features.

| Feature | Formulas |
|---|---|
| (Mean-ideal RPM) | $\frac{1}{N}\sum_{k=1}^{N} X_k - RPM_{ideal}$ |
| (Variance-ideal RPM) | $\frac{1}{N}\sum_{k=1}^{N} \left(\frac{X_k - m}{\sigma}\right)^2 - RPM_{ideal}$ |
| Skewness | $\frac{1}{N}\sum_{k=1}^{N} \left(\frac{X_k - m}{\sigma}\right)^3$ |
| Kurtosis | $\frac{1}{N}\sum_{k=1}^{N} \left(\frac{X_k - m}{\sigma}\right)^4$ |
| (Max-ideal RPM) | $\max(X_k) - RPM_{ideal}$ |
| (Min-ideal RPM) | $\min(X_k) - RPM_{ideal}$ |
| (Median-ideal RPM) | $X_{\frac{N+1}{2}} - RPM_{ideal}$ |

In the formulas, $X_k$ represents the instantaneous RPM data at time window $k$, $N$ denotes the total number of data points, $m$ represents the mean of the data, and $\sigma$ is the standard deviation of the data. $RPM_{ideal}$ refers to the engine governor command speed.

By extracting these time-domain features, which incorporated the differences from the ideal RPM, the bias in the learning process could be reduced, enabling the DNN model to effectively distinguish between abnormal and normal states.

### 4.2. Time-Domain Analysis

In this study, the aim was to address the potential degradation in performance when dealing with low-precision data. To investigate this, the precision of the original data was adjusted to simulate scenarios with varying precision conditions. The downsampling technique was employed to proportionally reduce the data precision. The downsampling rate, denoted as $n$, was varied to ($2\times$, $5\times$, $10\times$, $20\times$, $50\times$, $100\times$, and $200\times$) to examine the changes in the time-domain feature information extracted from the downsampled data, as well as the performance of the anomaly detection model utilizing these varied features.

Figure 9 illustrates the changes in the time-domain feature information as the downsampling rate increased compared to the original data. It can be observed that the time-domain features began to change slightly as the downsampling rate increased, but from a downsampling rate of $50\times$ the differences from the original data became more significant. This indicated that the time-domain feature information was highly sensitive to data precision.

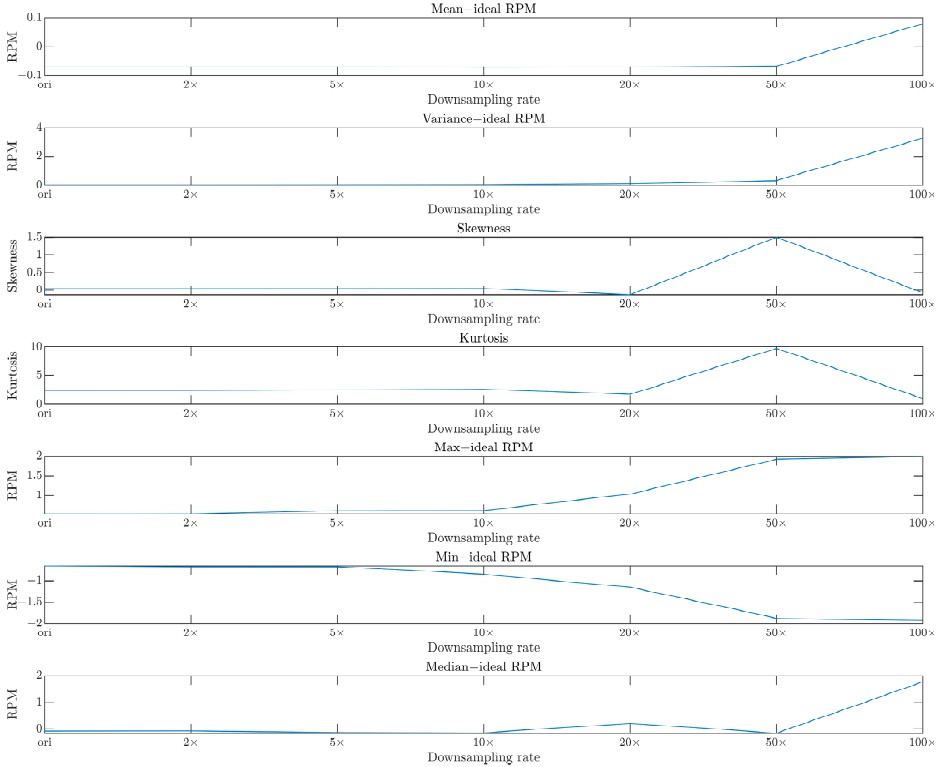

**Figure 9.** Time-domain features when varying the downsampling rate.

Figure 10 presents the fault detection accuracy in relation to the downsampling rate. The accuracy stood at 98.99% for the original data (at a downsampling rate of $1\times$), and even at a downsampling rate of $20\times$ the performance degradation remained at approximately 2–3% compared to the original data. However, beyond a downsampling rate of $50\times$, the accuracy saw a sharp decline only to about 60%.

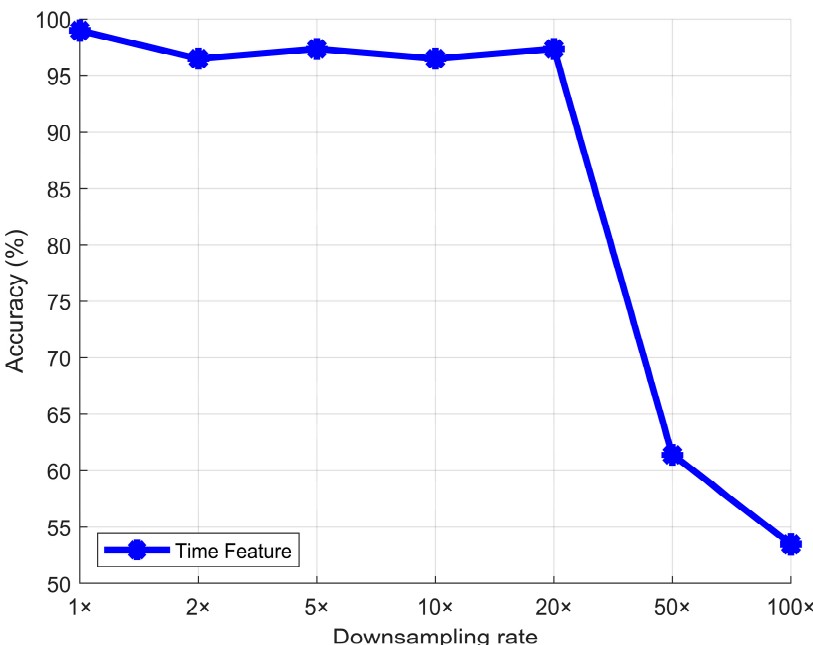

**Figure 10.** Fault detection accuracy in relation to the downsamping rate.

Considering the change in the time-domain feature information with the downsampling rate depicted in Figure 9, it could be concluded that the significant decrease in performance observed from a downsampling rate of 50× in Figure 10 was due to the time-domain features, under low-data precision, to effectively distinguish the features of anomalous data from normal data.

Based on these findings, it was determined that relying solely on time-domain feature information is insufficient to maintain satisfactory performance when the data precision is low. Therefore, in addition to the time domain, a combined analysis in the frequency domain was conducted to improve the anomaly detection performance.

## 5. Frequency-Domain Analysis

### 5.1. Data Preprocessing in the Frequency Domain

In addition to the time-domain analysis, another approach to extract feature information from the instantaneous RPM data was through frequency-domain analysis. The frequency domain allowed for the analysis of signal characteristics based on frequency components rather than changes over time, providing a different perspective on the RPM data. However, analyzing RPM frequencies can be challenging due to noise introduced by various factors, such as internal inertia, vibrations, and engine cylinder and crank impacts. Directly converting the signal into the frequency domain may result in distorted feature information.

To overcome this challenge, independent component analysis (ICA) was employed to extract significant features from the instantaneous RPM data, which contained multiple signal components and noise. ICA is a technique used to separate mixed signals into their original source components. It aims to find a demixing system that can recover the source signals from the observed mixed signals.

$$x = \begin{Bmatrix} x_1 \\ x_2 \\ \vdots \\ x_n \end{Bmatrix} \tag{4}$$

$$s = \begin{Bmatrix} s_1 \\ s_2 \\ \vdots \\ s_m \end{Bmatrix} \tag{5}$$

$$A = \begin{bmatrix} a_{11} & \cdots & a_{1m} \\ \vdots & \ddots & \vdots \\ a_{m1} & \cdots & a_{nm} \end{bmatrix} \tag{6}$$

Figure 11 illustrates the process for the ICA. In Figure 11, $x$ represents the discrete signal vector for the instantaneous RPM (Formula (4)), $s$ represents the discrete signal vector for the source (Formula (5)), and $A$ represents the mixing matrix that represents the correlation between the observation and the source (Formula (6)). The goal of the ICA was to find the demixing system $W$ (an $m \times n$ matrix) such that $s = Wx \cong s$, where s is statistically independent, allowing the extraction of the desired features. However, in this study, we encountered the problem of underdetermined blind source separation (BSS), since we observed $x$ through the single encoder sensor, resulting in a lower number of observations compared to the number of source signals. To address this problem, the combined method of empirical mode decomposition (EMD) and ICA, known as EMD-ICA, was utilized [36].

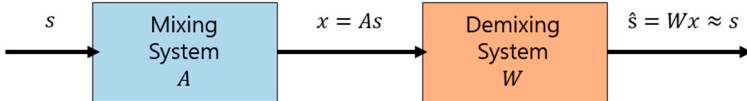

**Figure 11.** Process for the ICA.

EMD is an adaptive mode that decomposes a signal into intrinsic mode functions (IMFs), which are finite functions representing the different frequency components in the signal [37]. Figure 12 demonstrates the application of the EMD algorithm to decompose normal and abnormal instantaneous RPM data into six IMFs.

In this paper, the EMD-ICA approach was employed to solve the blind source separation problem caused by a mismatch between the number of observations ($x$) and the number of source signals (s). The EMD step was utilized to address the BSS problem before applying the ICA algorithm, allowing for the extraction of the main signal from the instantaneous RPM data, which contained multiple signal components and noise.

*5.2. Frequency-Domain Analysis Method*

When analyzing anomalies in the frequency domain, it was important to not only consider the magnitude of the signal but also its nonlinear features like the phase conjugation, which often occurred during anomalies. The power spectrum method, commonly used for frequency analysis, calculated the power distribution across frequencies, but disregarded the phase information and assumes linearity in the analysis.

To overcome these limitations, this paper employed the bispectrum method, which was effective in analyzing the nonlinear characteristics and captures the phase conjugation between the frequency components. Assuming that the EMD-ICA generated an instantaneous RPM signal represented by *X(t)*, we focused on exploring the frequency correlations within the *X(t)* signal by specifying a frequency range $f$ and identifying specific frequencies $f_1$ and $f_2$ within that range. The bispectrum formula $B_{xxx}(f_1, f_2)$ was defined by Formula (7).

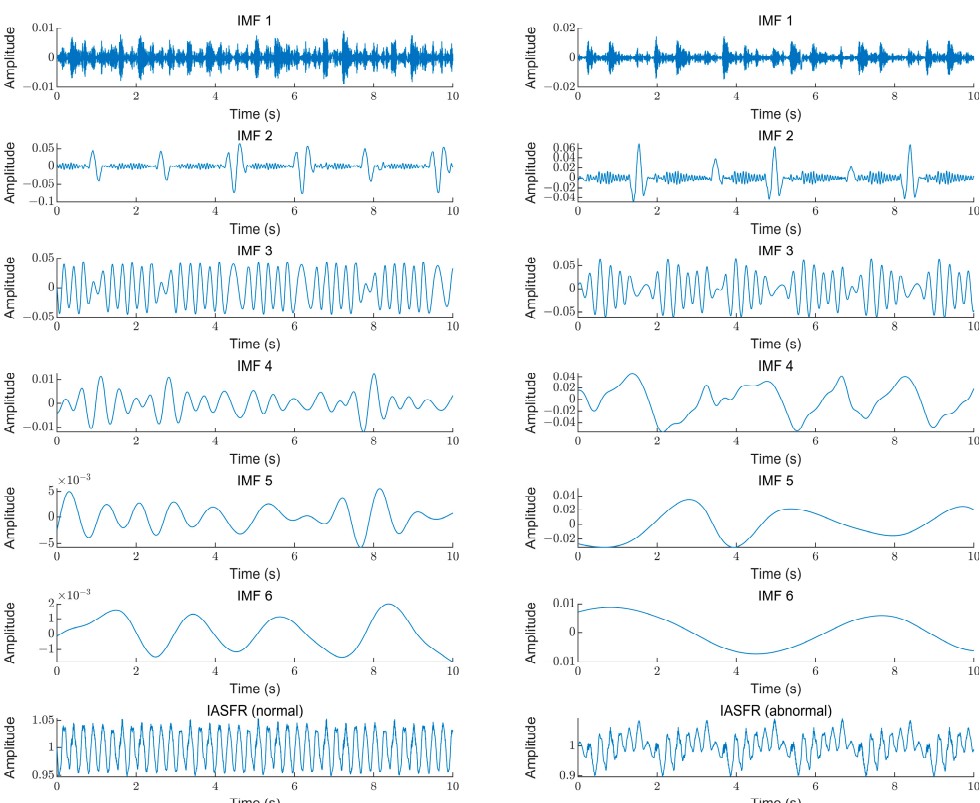

**Figure 12.** Results of the EMD (left: normal, right: abnormal).

$$B_{xxx}(f_1, f_2) = \lim_{T \to \infty} \frac{1}{T} E[X_T(f_1)X_T(f_2)X_T^*(f_1 + f_2)] \tag{7}$$

In this formula, "\*" represents the complex conjugation, and $X_T(f)$ denotes the Fourier coefficient. This formula revealed the correlation between not only the frequencies $f_1$ and $f_2$, but also the combination of $f_1$ and $f_2$. The amplitude and phase of $f_1$ and $f_2$ were zero except for the phase value of $(f_1 + f_2)$ [38].

By substituting $X(t)$ into Formula (7), a frequency spectrum was obtained, revealing the presence of phase-conjugated signals. This allowed for the identification of frequency regions where the conjugated phase occurred in the signal $X(t)$. Figure 13 presents the bispectrum analysis results from the autocorrelation of the instantaneous RPM data $X(t)$ used in conjunction with the EMD-ICA. To determine the magnitude of the bispectrum, the bispectrum results between the frequencies $f_1$ and $f_2$ were observed at an angle parallel to the *x*-axis (frequency $f_1$).

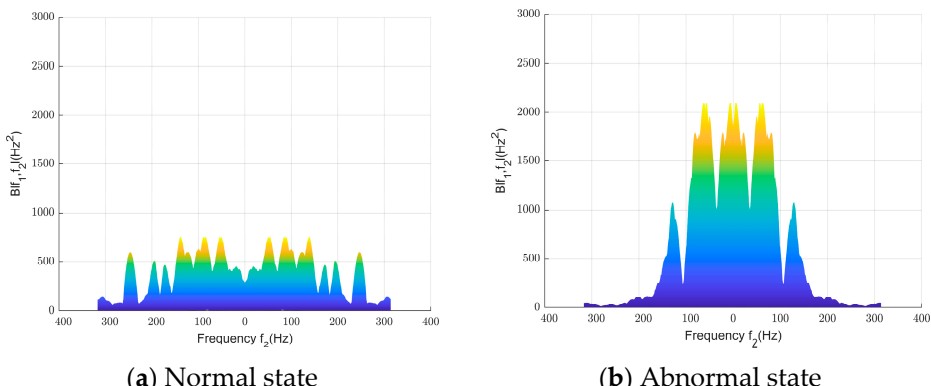

(**a**) Normal state          (**b**) Abnormal state

**Figure 13.** Bispectrum results.

Figure 13 illustrates the bispectrum results for the normal (Figure 13a) and abnormal (Figure 13b) states. In the bispectrum analysis, areas without phase conjugation exhibited values close to zero, while areas with phase conjugation had values greater than zero. The results indicated that the magnitudes in the bispectrum results for the anomalous data, where the phase conjugation was more likely to occur, were larger than those for normal cases.

These findings indicated that when an anomaly was present in the instantaneous RPM data, phase conjugation between the signals was more likely to occur. Since each value in the spectrum analysis results represented phase coupling, it can be used as a feature for anomaly detection.

### 5.3. Frequency-Domain Feature Extraction

In frequency-domain analysis, the bispectrum method was used to capture nonlinear features, such as the phase conjugation, for anomaly detection. However, the bispectrum result was a two-dimensional matrix in the form of an image, which can pose challenges when using it as input for training a DNN model. To address this issue, feature extraction techniques were employed to condense the bispectrum result into a more suitable format.

Figure 14 demonstrates the process for bispectrum feature extraction. The bispectrum result exhibited a symmetrical relationship, similar to the symmetry observed in the power spectrum. As a result, a triangular area within the bispectrum result, indicated by the bold lines in Figure 14, can be shifted to generate the remaining areas. Notably, the brighter areas within the triangular region represent significant correlations. This characteristic allowed for effective dimensionality reduction by either adding up the bispectrum values or selecting the maximum values by column within the region. These operations can be performed on the entire bispectrum result or on specific parts of it, confined to a limited area [39].

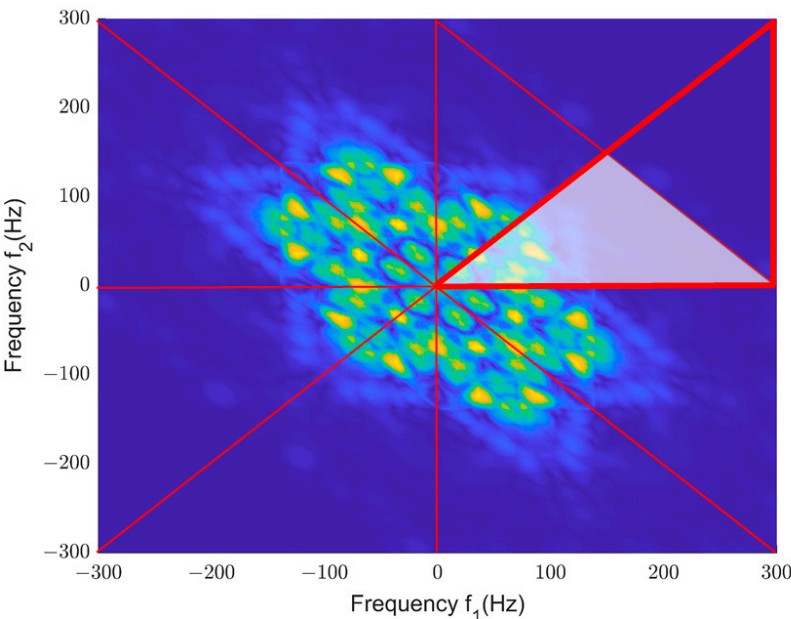

**Figure 14.** Bispectrum feature extraction. Whole figure comprised of bolded red line section, and a brighter area indicates where significant correlations occurred.

To extract feature information in the frequency domain, both methods of adding up the bispectrum values and selecting the maximum values from each column in brighter areas in the bispectrum image were utilized. This allowed for the extraction of key characteristics that represented the frequency features. By condensing the bispectrum result using these approaches, the dimensionality of the data was reduced, making it more amenable for input into a DNN model.

*5.4. Analysis in the Frequency Domain*

In the frequency-domain analysis, we conducted downsampling on the pulse data to simulate low-data precision. We then extracted the frequency-domain feature information and evaluated the performance of the anomaly detection model based on these features.

Figure 15 illustrates the performance changes in the DNN model for anomaly detection based on the frequency-domain feature information at different downsampling rates. The experimental setup details can be found in Section 6 of the paper.

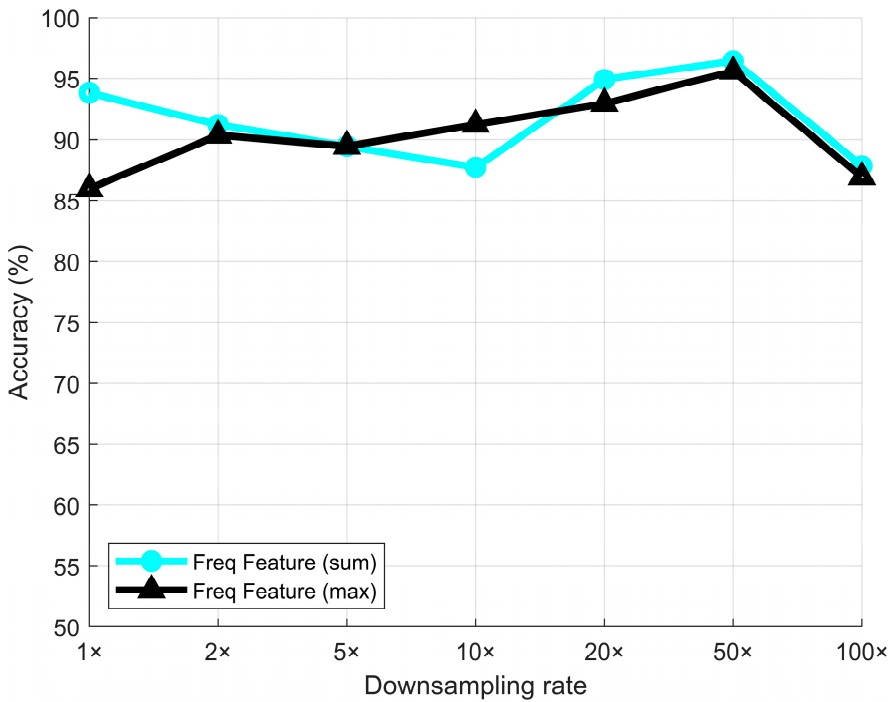

**Figure 15.** DNN performance changes at different downsampling rates (freq).

Analyzing the results, we observed that the sum of the frequency features (sum) consistently maintained a high accuracy of 90% throughout the downsampling rates. This indicated that even with decreasing data precision due to downsampling, the anomaly detection model based on the sum of the frequency features remained robust, with only a 10% deviation from the original accuracy.

Similarly, the maximum value of the frequency features (max) also exhibited a stable performance above 85% across all downsampling rates. This further demonstrated the model's resilience to low-data precision.

Overall, the results suggested that the anomaly detection model utilizing the frequency-domain feature information was capable of maintaining good performance even under degrading downsampling rates, ensuring its effectiveness in practical applications with low-precision data.

## 6. Results

*6.1. Dataset Configuration*

Table 5 summarizes the results of the experiments conducted in both the time and frequency domains. In the time-domain analysis, the average accuracy remained high at 97.3% from a downsampling rate of $1\times$ to $20\times$. However, a significant performance degradation occurred after the downsampling rate of $50\times$, and the average accuracy of the entire downsampling rate range decreased by 11.4% to 85.9%.

**Table 5.** Accuracy of the models in the time and frequency domains.

| Acc. (%)\DS Rate | 1× | 2× | 5× | 10× | 20× | 50× | 100× |
|---|---|---|---|---|---|---|---|
| Time | 98.99 | 96.49 | 97.37 | 96.49 | 97.37 | 61.4 | 53.48 |
| Freq (max) | 85.96 | 90.36 | 89.47 | 91.23 | 92.98 | 95.61 | 86.92 |
| Freq (sum) | 93.86 | 91.23 | 89.47 | 87.72 | 94.94 | 96.49 | 87.85 |

In the frequency domain, using the maximum value (max) method yielded an average accuracy of 90%, while the sum of the bispectrum values (sum) method resulted in an average accuracy of 91.4%, both until a downsampling rate of 20×. Although these results showed lower performance compared to the time-domain analysis, the frequency domain exhibited higher robustness to precision. For the entire downsampling rate range, the sum method yielded an average accuracy of 91.6%, and the max method resulted in an average accuracy of 90.4%.

The analysis results indicated that the time domain had high accuracy but low robustness to precision, while the frequency domain had relatively lower accuracy but higher robustness to precision. Since the two domains had complementary characteristics, their combination was expected to improve accuracy and robustness. Therefore, the paper proposed a time–frequency feature fusion approach that combined the features from both domains. Two time–frequency feature fusion datasets were created by combining the feature information extracted from the frequency domain (using the sum and max methods) with the time-domain feature information.

### 6.2. Experiment Results

In this section, we present the experimental results for the proposed feature fusion approach. To evaluate the performance, we implemented the DNN-based fault detection models using the time-domain features, the frequency-domain features, and the fused features combining the time and frequency information.

The model with the time-domain features can represent the existing algorithms that involve transforming the vibration data to RPM and extracting the statistical features [12]. On the other hand, the model employing the frequency-domain features represented the approach in [36], which classified the faults based on a support vector machine (SVM) with features extracted through the Wigner bispectrum.

In our analysis, domain-specific analysis and feature extraction were conducted in the MATLAB environment, while the training and prediction of the DNN model using the generated dataset were performed in the PyTorch environment. The results from the feature fusion, combining the time-domain analysis and frequency-domain analysis in a low-data precision environment, are presented in Figure 16 and Table 6. And in the process of the DNN training, validation loss and training loss with respect to the training epochs are presented in Figure 16.

Figure 17 presents the DNN model accuracy at various downsampling rates. The results, as shown in Figure 17 and Table 6, clearly demonstrate that the time–frequency feature fusion, which leveraged both time-domain and frequency-domain information, exhibited superior robustness to downsampling compared to using time-domain features alone, as well as outperforming the use of frequency-domain features alone.

Notably, the time–frequency feature fusion approach, combining time-domain feature information with features extracted using the sum method in the frequency domain, demonstrated exceptional performance across all downsampling rates, achieving 100% accuracy from downsampling rates from 1× to 10×. Even at higher downsampling rates exceeding 20×, the proposed model remained robust, maintaining an accuracy of over 96%. These results strongly indicated that the synergy between the time-domain and frequency-domain features leads to substantial performance improvements.

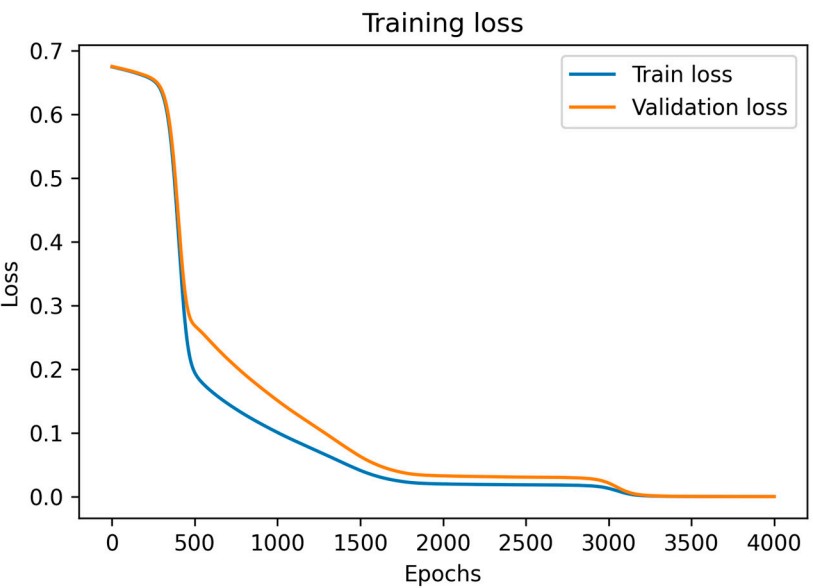

**Figure 16.** Loss graph for the DNN training process.

**Table 6.** Accuracy of the models using the proposed time freq feature fusion dataset.

| Acc. (%)\DS Rate | 1× | 2× | 5× | 10× | 20× | 50× | 100× |
|---|---|---|---|---|---|---|---|
| Time | 98.99 | 96.49 | 97.37 | 96.49 | 97.37 | 61.4 | 53.48 |
| Freq (max) | 85.96 | 90.36 | 89.47 | 91.23 | 92.98 | 95.61 | 86.92 |
| Freq (sum) | 93.86 | 91.23 | 89.47 | 87.72 | 94.94 | 96.49 | 87.85 |
| Time-Freq (sum) | 100.00 | 100.00 | 100.00 | 100.00 | 99.12 | 99.12 | 96.26 |

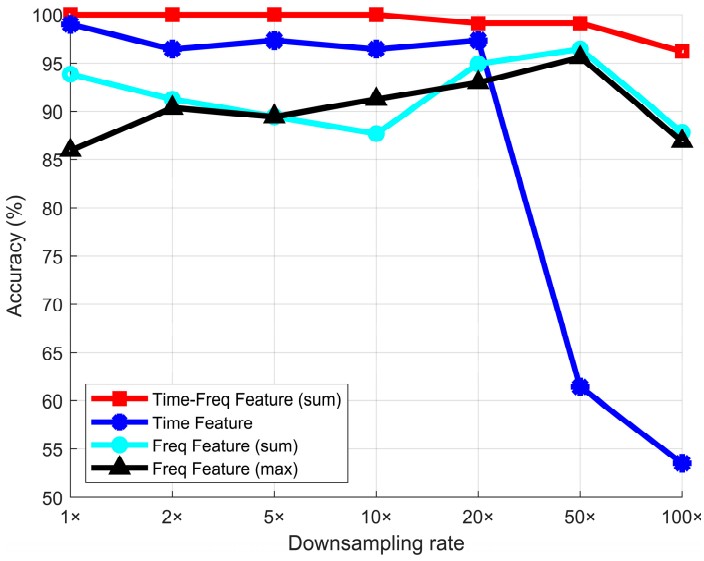

**Figure 17.** DNN performance changes at different downsampling rates (total).

In summary, combining features from both the time and frequency domains through time–frequency feature fusion enhanced the accuracy and robustness of the analysis. The proposed approach proved to be a promising method for achieving improved performance compared to using features solely from the time or frequency domain.

## 7. Conclusions

In this research, we introduced the time–frequency feature fusion method, which combined statistical feature information in the time domain with bispectrum based feature information in the frequency domain, based on instantaneous RPM. Our experiments demonstrated that the time–frequency feature fusion method, which integrated the feature information from both domains, outperformed the individual analysis methods in all scenarios, particularly with the original data, where it achieved higher accuracy than analyzing only the time domain.

These findings highlight the robustness of the proposed approach to low-data precision, as it maintained high performance even with imprecise data. This method can be particularly valuable in situations where equipment limitations result in the collection of low-precision data. By utilizing the time–frequency feature fusion method, the accuracy of anomaly diagnosis predictions can be improved even under such conditions.

For future work, it would be valuable to investigate the development of fault detection methods under voyage environmental conditions. This could involve adapting the proposed approach to handle real-world scenarios and account for the complexities and challenges associated with operating in a dynamic environment.

**Author Contributions:** Conceptualization, H.J.-G., S.-J.L., H.-S.L., J.-H.Y. (Jung-Hee Yang) and S.K.; methodology, H.J.-G., S.-J.L. and H.-S.L.; software, H.J.-G. and S.-J.L.; validation, J.-H.Y. (Jung-Hee Yang) and S.K.; formal analysis, J.-H.Y. (Jung-Hee Yang) and S.K.; investigation, H.J.-G., S.-J.L. and J.-H.Y. (Jeong-Hyun Yoon); resources, H.-S.L., J.-H.Y. (Jung-Hee Yang) and S.K.; data curation, J.-H.Y. (Jung-Hee Yang) and S.K.; writing—original draft preparation, H.J.-G., S.-J.L., J.-H.Y. (Jeong-Hyun Yoon), H.-S.L. and S.K.; writing—review and editing, H.-S.L. and S.K.; visualization, H.J.-G. and S.-J.L.; supervision, H.-S.L. and S.K.; project administration, J.-H.Y. (Jung-Hee Yang) and S.K.; funding acquisition, J.-H.Y. (Jung-Hee Yang) and S.K. All authors have read and agreed to the published version of the manuscript.

**Funding:** This research has received partial support from the MSIT (Ministry of Science and ICT), Republic of Korea, under the ITRC (Information Technology Research Center) support program (IITP-2023-2021-0-01816), supervised by the IITP (Institute for Information and Communications Technology, Planning and Evaluation) (50%). Additionally, it has been funded by the MSIT, Republic of Korea, through the ICAN (ICT Challenge and Advanced Network of HRD) program (IITP-2023-RS-2022-00156345), also supervised by the IITP (50%).

**Institutional Review Board Statement:** Not applicable.

**Informed Consent Statement:** Not applicable.

**Data Availability Statement:** Data sharing is not applicable to this article due to privacy and security issues.

**Conflicts of Interest:** The authors declare no conflict of interest.

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
