# Peer review of "A Novel Time–Frequency Feature Fusion Approach for Robust Fault Detection in a Marine Main Engine"

_jmse, doi:10.3390/jmse11081577_

Round 1
Reviewer 1 Report
1- The abstract does not replicate what is conducted in the paper adequately.
2- The obtained results need to be reflected by values in the abstract.
3- Some abbreviations, such as RPM in the abstract, are presented without prior detention or defined later; please check them.
4- Reduce the used keywords to make them at most 5.
5- A short paragraph showing how the paper is organized essential to be presented at the end of the introduction section.
6- Please give more explanation to the following sentence, which appeared in the introduction section “However, these time-domain statistical features rely on real-time values, which may pose challenges in accurately extracting features when the data contains faults.”
7- The title of Figure 2 is not informative enough to convey its content. Please provide a more descriptive and clear title.
8- Figure 3 is unclear, especially the sensor picture; please, present a better figure.
9- The information presented in Table 1 is not dimensions, as stated in the paragraph directly below this table; please correct that.
10- Why was only one fault type considered in this research?
11- The title of the verticle axis in Figure 4 must be presented in the figure.
12- Please enhance the resolution of Figures 4, 5, 8, and 11.
13- Please refer to the software used to perform the proposed approach.
14- It is advised that Section 3 comes after Section 6; however, Subsection 3.2 should be a separate section that comes before the Time Domain section.
15- The Data Processing subsection is not easy to understand; please, put more effort to enhance it.
16- There is an obvious leakage of DNN results in the article, so the paper cannot be accepted in its current state.
The manuscript has some typos and grammatical errors. Thus, it is strongly recommended that the whole article be proofread carefully.
Author Response
The authors would like to express sincere gratitude for your dedicated efforts in reviewing this paper. The authors greatly appreciate your insightful comments, which have proven to be invaluable in enhancing the overall quality of this work. Please see the attachment. Thank you

Reviewer 2 Report
This paper discusses vibration data-based fault detection in marine main engines. The paper is interesting, but the writing could be clearer and more cohesive. Significant modification needs to be done to this paper. Additionally, the reviewer has the following question and comments that need to be addressed and incorporated into the modified version of this paper.
- The literature review is not thoroughly done. Only 14 papers are not enough. As this is not such a novel study, an exhaustive literature review is required and also mentions their drawbacks in this paper. A standard journal paper contains 40-50 references.
- The proposed framework is confusing. Please show a flowchart to demonstrate the end-to-end framework about what is the input and output of the framework properly.
- The figure xlabel, ylabel, legends, and all the figure writings are not of the correct size and hence not properly visible. Ensure that the figure fonts are as big as the text fonts. Look at Figs 4 and 5 especially.
- In Table 2, the DNN architecture is shown. Please mention the reason behind choosing this ANN. Why not CNN? Why 7 hidden layer? Why this number of nodes? Is it optimally tuned? Show hyperparameter tuning study. Is this generalized or only trained for this machine?
- Also, discuss the process of data generation properly, is everything trained on real data? How did you label the training data?
- Page 6 Line 173, what does the word mean “quay”?
- How fast is the deep learning inference? Can this framework be applied to real-time health monitoring of engines?
- What is the reason for dropping the accuracy beyond 20x in Fig 9. In this paper, only it is mentioned that accuracy drops beyond 20x, the reason is not mentioned.
- In Eq. 3 and 4, use proper brackets (curly) maybe to denote the ‘x’ and ‘s’ are vectors
- Page 11, line 316, 1 and 2 in f1 and f2 would be subscripted. Please modify that.
- In fig 12, why the abnormal state, the signal energy is only focused in the lower frequency? And why such a wavy pattern in the normal state as well?
- Table 5 shows the fusion of time and frequency characteristics. Please state the reason why it is happening. This means that in some cases, the accuracies are better, and in other cases, it is not.
- Did the authors think about combining the time, freq (max), and freq (sum) features together to boost the overall accuracy?
- Please show a comparison with state-of-the-art literature. I am sure a lot of papers would address machine health from vibration data.
- Please consider referring to the following related papers.
- Xi, W., Li, Z., Tian, Z., & Duan, Z. (2018). A feature extraction and visualization method for fault detection of marine diesel engines. Measurement, 116, 429-437.
- Jana, D., Nagarajaiah, S., Yang, Y. and Li, S., 2021. Real-time cable tension estimation from acceleration measurements using wireless sensors with packet data losses: Analytics with compressive sensing and sparse component analysis. Journal of Civil Structural Health Monitoring, pp.1-19.
- Li, Z., Yan, X., Guo, Z., Zhang, Y., Yuan, C., & Peng, Z. (2012). Condition monitoring and fault diagnosis for marine diesel engines using information fusion techniques. Elektronika ir Elektrotechnika, 123(7), 109-112.
- Jana, D., Patil, J., Herkal, S., Nagarajaiah, S. and Duenas-Osorio, L., 2022. CNN and Convolutional Autoencoder (CAE) based real-time sensor fault detection, localization, and correction. Mechanical Systems and Signal Processing, 169, p.108723.
- Yan, G., Hu, Y., & Jiang, J. (2022). A novel fault diagnosis method for marine blower with vibration signals. Polish Maritime Research, 29(2), 77-86.
Please thoroughly check the typographical and grammatical errors before paper submission.
Author Response

(The authors gave the same response as above.)

Reviewer 3 Report
This manuscript takes the marine main engine as the research object, and proposes an anomaly detection model for the marine main engine based on time-frequency feature fusion under the condition of low data accuracy, and verifies the effectiveness of the proposed method by comparing with separate time-domain and frequency-domain features. The main problems are as follows:
(1) The title of this manuscript should be revised. Deep neural networks are mentioned in the title, but there is little introduction to deep neural networks in the text, and the title should be related to the Time-Frequency Feature Fusion and low data accuracy that this manuscript focuses on.
(2) The research object should be unified, and the object introduced in the first paragraph of the Introduction section is internal combustion engines, and the content introduction is main engines.
(3) The fifth paragraph of the Introduction section only briefly lists the relevant literature of ship anomaly detection, and should summarize the listed literature, explain the current problems of marine anomaly detection and the improvements made in this article.
(4) Please verify the correctness of Equation (1) in the manuscript, and describe in detail how to convert the pulse signal into an instantaneous RPM signal by Equation (1).
(5) Data precision is mentioned several times in this manuscript, and a quantitative formula description of data accuracy should be given.
(6) The data was divided into 8 groups according to precision in section 3.2, and only 7 groups were used in sections 4.2 and 5.4.
(7) Section 5.2 describes "By applying Formula (6) to calculate X(t)", please explain how to calculate X(t) from Equation (6).
(8) The meaning and units of horizontal and vertical coordinates should be added in Figures 8 and 11, and units should be added to the horizontal and vertical coordinates of Figures 4, 5, 6, 9 and 11.
The manuscript has a low level of English writing, with a certain amount of incoherent sentences and some grammatical errors, which the author needs to carefully check and revise.
Author Response

(The authors gave the same response as above.)

Round 2
Reviewer 1 Report
Thanks for accurately considering all the given comments; however, it has been stated on the cover that a loss graph is added to the revised version, but I could see it in the paper! Please check.
Used English is fine just minor errors need to be corrected.
Author Response
Thank you for your thoughtful comments. We added the attachment below.

Reviewer 2 Report
The authors have addressed and incorporated all the comments. This version of the manuscript can be accepted for publication.
Author Response
We appreciate your efforts put into reviewing this manuscript. Thank you profusely.
Reviewer 3 Report
The manuscript has undergone a certain number of revisions based on the revisions, which solves most of the problems raised, but there are still some problems. The existing problems are as follows:
(1) The abstract is too long and needs to be concise, and the innovations in this article should be emphasized.
(2) The literature review part should not only briefly list the literature, but should lead to vibration signals from the problems existing in acoustic signal anomaly detection;Section 1.2.3 still includes acoustic and vibration signals when describing other sources;The relationship between the paragraphs in section 1.2.3 is not much different, and the division of paragraphs is based on?Line 170-194 in manuscript should be a separate section;It is necessary to increase the research status of anomaly detection under low-precision conditions.
(3) Figure 6 has an error, and the method proposed in this article is to enter the fused data into the DNN.
(4) Figure 7 has an error, Kurtosis is a dimensionless indicator.
(5) Some plots still have no units added to the horizontal and vertical coordinates. For example, Figure 9, 10, 12, 15.
The manuscript has a low level of English writing, with a certain amount of incoherent sentences and some grammatical errors, which the author needs to carefully check and revise.
Author Response

(The authors gave the same response as above.)

Round 3
Reviewer 3 Report
This manuscript has been modified to some extent in the light of the revisions, addressing some of the issues raised. The current problems are as follows:
(1) Section 1.2.1 still does not list the problems with anomaly detection using acoustic signals. The author may have misunderstood the question posed last time, "but should lead to vibration signals from the problems existing in acoustic signal anomaly detection" does not mean to add the literature on the use of vibration signals for fault detection in section 1.2.1, the use of vibration signals for anomaly detection also faces the problem of noise, and the author needs to list the advantages of using vibration signals over acoustic signals.
(2) In Section 1.2.3, the literature listed in line 119-156 in manuscript is not relevant to the title of this section. And when describing the relevant literature, it is necessary to add what signal source to use for detection, so as to correspond to the topic of this section.
In general, the manuscript has been revised and the content has been somewhat fleshed out, but there are still certain problems in the literature review section.
The level of English writing in this manuscript has improved somewhat, but there is still room for improvement.
Author Response
Please find the attached document for your review. While the authors have carefully considered seeking professional English editing from qualified experts, certain funding and logistical challenges arose during the revision process, limiting our access to English editing services. If the current revised version does not meet your expected standards, kindly recommend that we obtain professional editing. We sincerely value your continued interest in our study and are grateful for your comprehensive review.
